# Global declines in net primary production in the ocean color era

Greg M. Silsbe [1,4] ✉, James Fox[2,4] ✉, Toby K. Westberry [3] & Kimberly H. Halsey[2]

The majority of heat associated with climate change has been absorbed in the sunlit surface ocean where phytoplankton carry out half of biospheric net primary production (NPP). The physical entrainment of nutrients from depth into the surface constrains NPP across most of the ocean, therefore it has been widely hypothesized that a warmer and more thermally stratified ocean will diminish NPP. As phytoplankton are the dominant driver of ocean color, the satellite remote sensing record is the best approach to assess global NPP trends. Here we show that statistically significant decreases in NPP have occurred in almost half of the ocean and these changes are dominated by declines in the tropical and subtropical stratified ocean. A deeper analysis confirms that strengthening nutrient limitation is largely driving declining NPP. Climate-mediated shifts in NPP represent a fundamental perturbation to biogeochemical cycles that can further weaken global fisheries.

The ocean has absorbed more than 90% of the heat generated by anthropogenic climate change, resulting in a warmer and more thermally stratified environment across the vast majority of the ocean[1,2]. Most of this heat is confined to the upper few hundred meters of the ocean[2], and it is in these environments where phytoplankton, a diverse assemblage of microscopic protists and cyanobacteria, carry out approximately half of biospheric photosynthesis[3]. Phytoplankton net primary production (NPP), the rate at which organic carbon is generated through photosynthesis, sets the foundation for marine food webs and fisheries[4], is a key component of global biogeochemical cycles and a primary driver of biogenic carbon sequestration to the deep ocean[5]. The physical entrainment of vital nutrients from depth into the well-lit surface layer constrains NPP in most of the ocean[6,7]. Therefore, the prevailing conceptual view is that a warmer ocean with stronger thermal gradients will weaken nutrient entrainment and diminish NPP in much of the ocean[8,9]. Yet, how climate change is impacting NPP is still unclear. For example, multi-decadal direct measurements in the Atlantic and Pacific Oceans[10,11] and earth system model projections show opposing trends[12].

The key role of NPP in ecosystem services and the global carbon cycle has given rise to an ongoing pursuit of accurate, synoptic estimates at high resolution in both time and space[13–15]. Satellite remote sensing is the only platform that offers this capability, and past and present ocean color missions collectively provide an unbroken global time. Continual advances algorithms derived from remote sensing reflectance ($R_{rs}$), the primary metric of ocean color, have fundamentally improved the way we observe and estimate phytoplankton biomass, physiology, growth rates, and ultimately NPP from space[16–18].

Here we present a 25-year global assessment across two satellite missions (SeaWiFS and MODIS-Aqua) to investigate how a warming ocean has influenced NPP in conjunction with underlying metrics of nutrient limitation. We use an absorption-based model that derives NPP as the product of photosynthetic active radiation (PAR), the fraction of PAR absorbed by phytoplankton ($a_\phi/a$, where $Q_{PAR} = PAR \times a_\phi/a$), and the efficiency with which absorbed photosynthetic energy is converted into carbon biomass ($\phi_\mu$)[19]. Direct field measurements[20,21] and culture-based experiments[22,23] have shown that $\phi_\mu$ has muted variance across taxonomic and nutrient gradients, consequently the vast majority of NPP variance is governed by ocean color measurements of $Q_{PAR}$. Physiological responses to increased nutrient limitation in phytoplankton are well understood and visible in

[1]Horn Point Laboratory, University of Maryland Center for Environmental Science, Cambridge, MD, USA. [2]Department of Microbiology, Oregon State University, Corvallis, OR, USA. [3]Department of Botany and Plant Pathology, Oregon State University, Corvallis, OR, USA. [4]These authors contributed equally: Greg M. Silsbe, James Fox. ✉e-mail: gsilsbe@umces.edu; james.fox@oregonstate.edu

the ocean color record[24]. As nutrient limitation increases, phytoplankton manufacture fewer light-harvesting pigments and associated photosynthetic machinery to match the lower energetic demands for growth[25,26]. This phenomenon is observable as concomitant decreases in the phytoplankton pigment absorption coefficient ($a_\phi$) and phytoplankton growth rates ($\mu$), where $\mu$ is derived by normalizing NPP to phytoplankton carbon biomass ($C_{Phyto}$) as estimated from particulate backscattering[27].

Monthly time series of NPP, growth rates, and all other ocean color parameters presented here ($n = 303$, global 1/12° data) were deseasonalized using monthly climatologies (1997–2022). Spatially explicit trend analysis (Type 1 ordinary least-squares regression and uncertainty analysis accounting for autocorrelation, see Methods), was performed on deseasonalized data and is presented as absolute trends and percent change relative to the 25-year mean. Empirical orthogonal function (EOF) analysis was performed to identify dominant modes of variance in time and space and compared to a suite of climate indices. The specific NPP model variant chosen has no direct dependence on sea surface temperature (SST)[19], allowing for an independent assessment of NPP trends and variance in a warming ocean.

## Results

### Trend analysis

Across the satellite record, NPP is decreasing throughout vast regions of the subtropical and tropical ocean and increasing in polar and subpolar waters (Fig. 1a). Overall, NPP has significantly declined in 48% of the ocean at an average rate of $-34$ mg C m$^{-2}$ d$^{-1}$ decade$^{-1}$, equivalent to a $-7.1$% normalized decadal rate. Conversely only 4% of the ocean has experienced a significant increase in NPP but at a higher average rate of 48 mg C m$^{-2}$ d$^{-1}$ decade$^{-1}$, this is equivalent to a $+20$% normalized decadal rate reflecting the fact that significant increases in NPP are largely occurring in low productivity regions (Fig. 1b, c). Spatially contiguous trends broadly correspond to major ocean biomes and currents (Supplementary Fig. 1). NPP is declining in all subtropical gyres and along currents that advect warm water poleward (Brazil, Gulf Stream, Kuroshio, Norwegian). Conversely, NPP is increasing along the austral subtropical front and either increasing or unchanged along eastern boundary currents that advect colder water towards the equator (California, Greenland, Malvinas, Oyashio, and Peru). A clear latitudinal anomaly is increasing NPP centered around 15°S in the southeast Pacific Ocean (Peru current). This region, along with the North Atlantic where NPP is also increasing, is one of the few areas globally where SST has cooled across the ocean color record[1]. The strong correspondence of SST and ΔNPP is further shown by grouping trends into respective 1 °C mean SST (1998–2022) bins and computing the area within each bin where NPP trends are significant (Fig. 1d, e). These data show that the 15 °C isotherm, which has been used to delineate temperate seasonally mixing biomes from warmer permanently stratified biomes[24], also represents the transition from increasing to decreasing ΔNPP (Fig. 1d). Above 15 °C most declining trends exceed the 90% confidence interval, whereas below this temperature most of the increasing trends are not significant, and the largest increases in NPP are occurring in polar waters (Fig. 1e).

Trend analyses of the atmospheric (PAR), bio-optical ($a_\phi$/a), and physiological ($\phi_\mu$) components of NPP demonstrates that changes in the fraction of energy absorbed by phytoplankton ($a_\phi$/a) is largely driving NPP trends (Fig. 2). Overall, $a_\phi$/a has significantly decreased in 43% of the ocean at an average normalized decadal rate of $-7.6$% and increased in 6% of the ocean at an average normalized decadal rate of $+18$%. Significant trends in PAR and $\phi_\mu$ are present in only 16 and 22% of the ocean, respectively, and are comparatively muted, with average decadal rates less than 4% (both positive and negative for each parameter). There is strong spatial overlap in regions where both NPP and $a_\phi$/a have significant trends. 86% (81%) of regions where NPP is significantly decreasing (increasing) has seen commensurate significant

trends in $a_\phi$/a. This analysis reveals that the dominant driver of NPP is embedded within the ocean color record, and the computationally simple term $a_\phi$/a, derived directly from satellite data, is a robust indicator of shifts in marine productivity.

Declining NPP and $a_\phi$/a across most of the stratified ocean is consistent with stronger and more pervasive nutrient limitation. However, these declines could also occur through nutrient-independent increases in the spectral absorption coefficient of dissolved organic matter ($a_{dg}$) or decreases in phytoplankton biomass ($C_{phyto}$) without commensurate changes in growth rates ($\mu$). Therefore to better resolve the mechanisms causing changes in NPP embedded in the ocean color record, normalized trends, slopes divided by their respective mean, of $a_\phi$ and $a_{dg}$, as well as euphotic zone integrated $C_{phyto}$ and $\mu$ were derived and directly compared. Significant decreases (increases) in $a_\phi$ are occurring in 36% (6%) of the ocean (Fig. 3), significant decreases (increases) in $a_{dg}$ are occurring in 25% (3%) of the ocean, a direct comparison of normalized decadal trends confirms that declining $a_\phi$ is the globally dominant trend in 33% of the ocean (Supplementary Fig. 2). Significant decreases (increases) in $\mu$ are occurring in 44% (4%) of the ocean and euphotic zone integrated $C_{phyto}$ also registers statistically significant decreases (increases) in 17% (6%) of the ocean. Moreover, the normalized decadal $C_{phyto}$ rates ($-2$% and $+4$%) are comparatively muted relative to the normalized decadal $\mu$ rates ($-11$%, $+23$%, Fig. 3). In this framework, decreases in $a_\phi$, $\mu$ and NPP strongly suggest that the warming and more thermally stratified surface ocean is an increasingly nutrient limited environment, and that cellular reductions in photosynthetic pigments rather than changes in biomass is the dominant response observable in the ocean color record. This observation is mathematically consistent with statistically significant changes in $R_{rs}$ that have emerged in more than half of the global ocean[28].

### NPP anomaly variability in space and time

Previous analyses of global NPP anomalies (1997–2008) have shown that interannual variability is on the order of ±2 Petagram (Pg) C year$^{-1}$ and is strongly influenced by SST ($r = 0.80$), particularly in the permanently stratified surface ocean[29]. Past analyses should be interpreted with some caution, however, as they often employ NPP models with first-order dependencies on SST[30] and, unlike the model used here, their behavior cannot be evaluated independently from changes in SST. Across the 25-year record, NPP averaged 56.8 Pg C year$^{-1}$ with a standard deviation of 1.9 Pg C year$^{-1}$ (Fig. 4a). Global NPP declined at an average rate of 1.8 Pg C decade$^{-1}$, though interannual variability is pervasive. This change represents a net 7.4% decrease over the satellite record, which greatly exceeds the CMIP6 high emissions scenario models (RCP8.5) that project NPP to decrease by 2.99% by the end of the 21st century[12].

To better elucidate the drivers and global patterns of NPP anomalies, we conducted EOF analysis to identify the dominant modes of NPP anomaly variance in space and time and then compared these modes to a suite of oceanic and atmospheric climate indices, including monthly global SST anomalies over the same time period (Fig. 4b). The first mode of global NPP anomaly variance (PC$_1$) explains 13.3% of total variance, encapsulates the largest magnitude of NPP variance with a total range of 2.8 Pg C, and corresponds to global declines in NPP that become increasingly pronounced towards the equator. Global SST anomalies were the highest correlated climate index ($r = -0.63$), suggesting that a warming ocean is now the dominant driver of interannual variability in marine photosynthesis. The second mode of variance explains 5.3% of variability, is weakly positively correlated with SST anomalies ($r = 0.23$) and encapsulates a comparatively smaller range of NPP (0.7 Pg C). This second mode has a strong latitudinal signal, with modest negative and positive NPP anomalies in the subtropical northern and southern hemispheres, respectively. The most striking feature of PC$_2$ is the pseudo-seasonal time component that emerges from deseasonalized data. A more detailed inspection of this

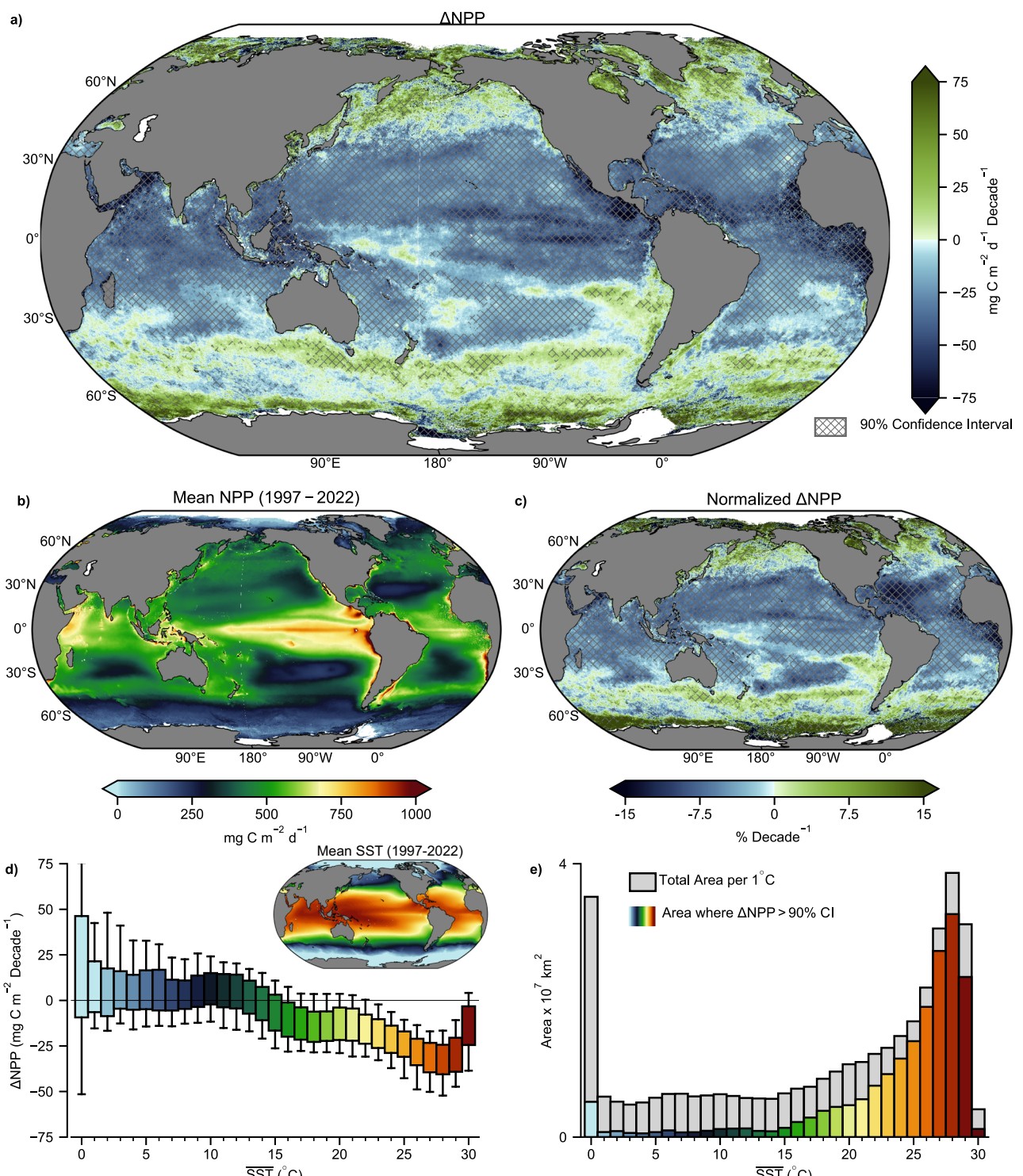

**Fig. 1 | Global trends in Net Primary Production (NPP). a** NPP anomaly trends (Sep-1997 to Dec-2022) where hatching is used to denote regions where the trend exceeds the 90% confidence interval. **b** Mean NPP across the same period and **c** trends expressed as percent change from the mean. **d** All NPP trends divided into respective 1 °C mean sea-surface temperature (SST) bins (1998–2022 average) shown in the inset where boxes represent the interquartile range and whiskers represent the 10 and 90th percentile. **e** the areal extent of significant NPP trends (colored) within each SST bin relative to the total ocean area within each SST bin.

mode shows that regions of high PC₂ variance occur near the 15 °C isotherm in both hemispheres and these regions have undergone pronounced phenological shifts in NPP (Supplementary Fig. 3). In regions of high $PC_2$ variance, NPP is increasing in the winter and decreasing in the summer in both hemispheres, resulting in diminished seasonality (Supplementary Fig. 3) that gives rise to pseudo-seasonal anomaly variance. That shifting phenology explains more of

the global NPP anomaly variance than other established climate indices (e.g., El Niño) is notable. The third mode of variance explains 3.9% of NPP anomaly variability, encapsulates a total range of 0.9 Pg C, and is highly negatively correlated with the Pacific Decadal Oscillation (PDO, $r = -0.63$). The transition from a positive to strongly negative PDO in part explains the reversal in declining global rates from 2018 to 2021. Finally, the fourth mode of variance explains 2.8% of variability,

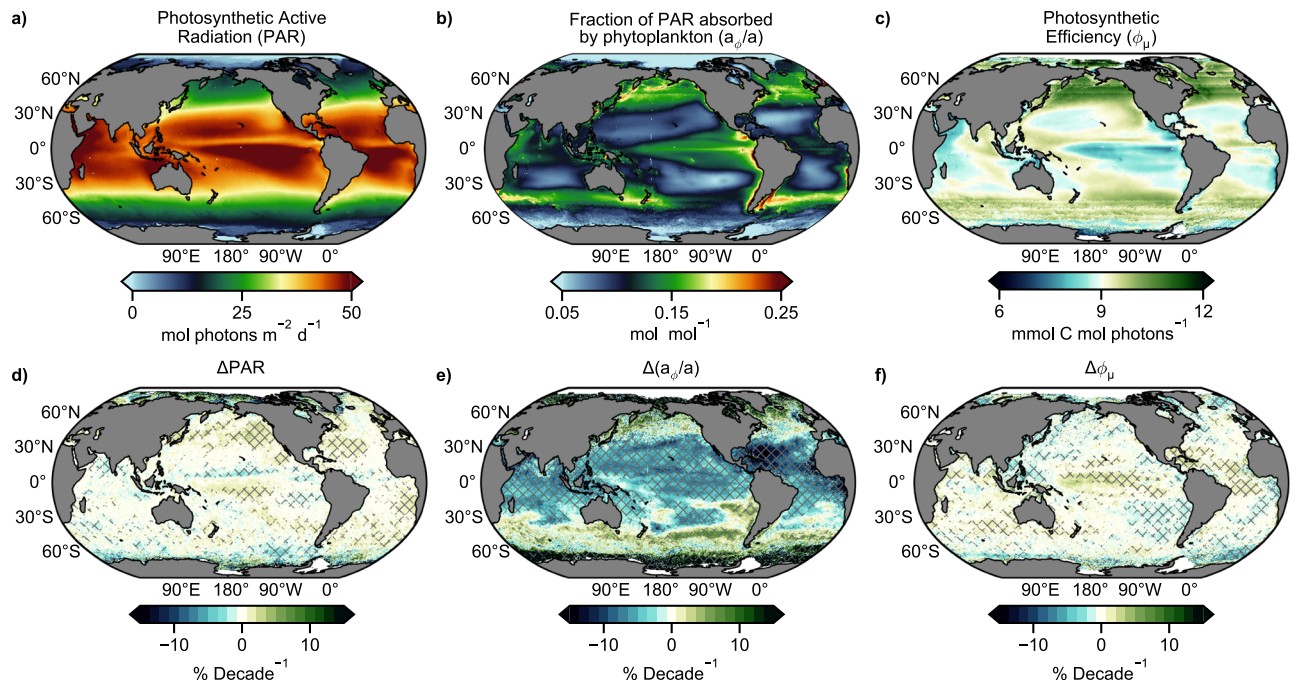

**Fig. 2 | Global patterns and trends in the atmospheric, bio-optical and physiological drivers of Net Primary Production.** Average **a** photosynthetic active radiation (PAR), **b** fraction of PAR absorbed by phytoplankton ($a_\phi/a$), and **c** net photosynthetic carbon efficiency ($\phi_\mu$), **d–f**, corresponding decadal trends normalized to their respective mean (1998–2022), where hatching denotes regions where the trend exceeds the 90% confidence interval.

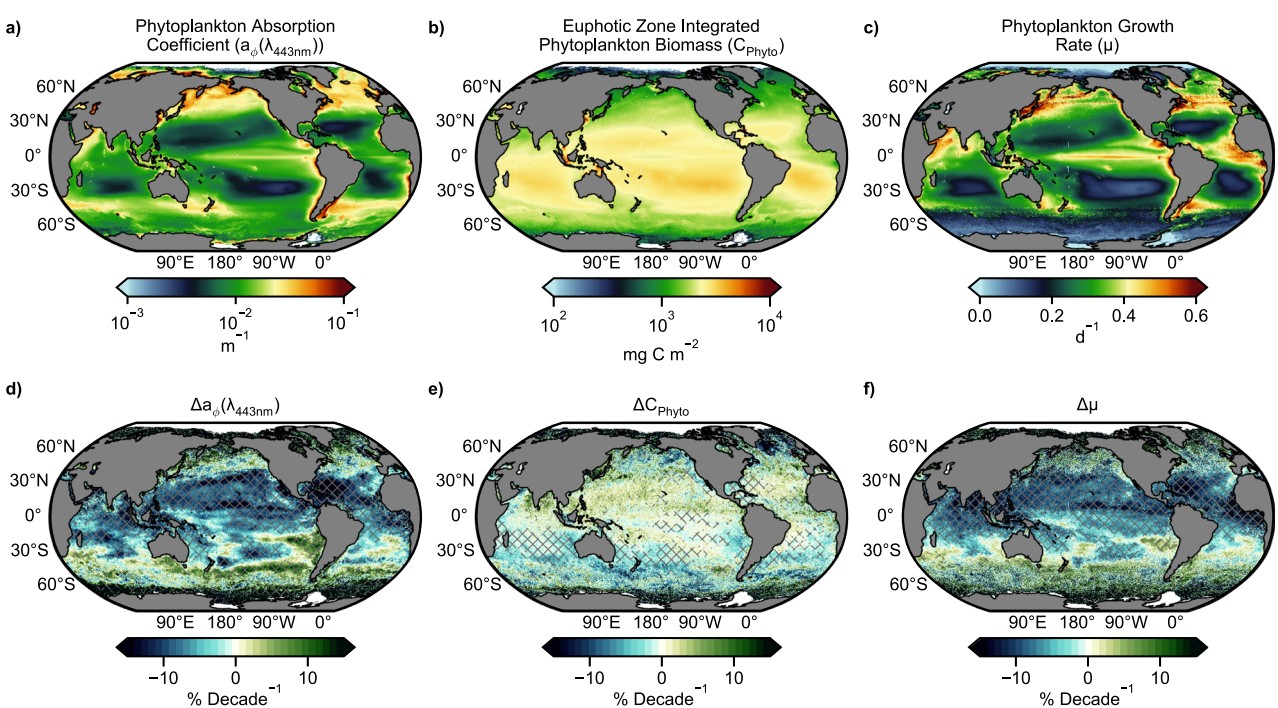

**Fig. 3 | Global patterns and trends in phytoplankton absorption coefficients, carbon biomass, and growth rates.** Average **a** phytoplankton absorption coefficient ($a_\phi$), **b** phytoplankton biomass integrated over the euphotic zone (Cphyto), and **c** phytoplankton growth rates (μ), **d–f** corresponding decadal trends normalized to their respective mean (1998–2022), where hatching denotes regions where the trend exceeds the 90% confidence interval.

encapsulates a total range of 0.5 Pg C, and is positively correlated with the El Niño 3.4 index (*r = 0.61*).

## Deepening of NPP

Satellite ocean color measurements are restricted to the upper few meters of the surface mixed layer. In most of the ocean, the euphotic zone (depths where NPP > 0) extends beyond the surface mixed layer, and phytoplankton residing at depths beyond the reach of ocean color sensors often form so-called deep chlorophyll maxima (DCM). A key feature of DCMs is that they often represent a photoacclimative response to low-light, whereby cellular pigmentation increases with no change in biomass[31]. The NPP model employed here includes a

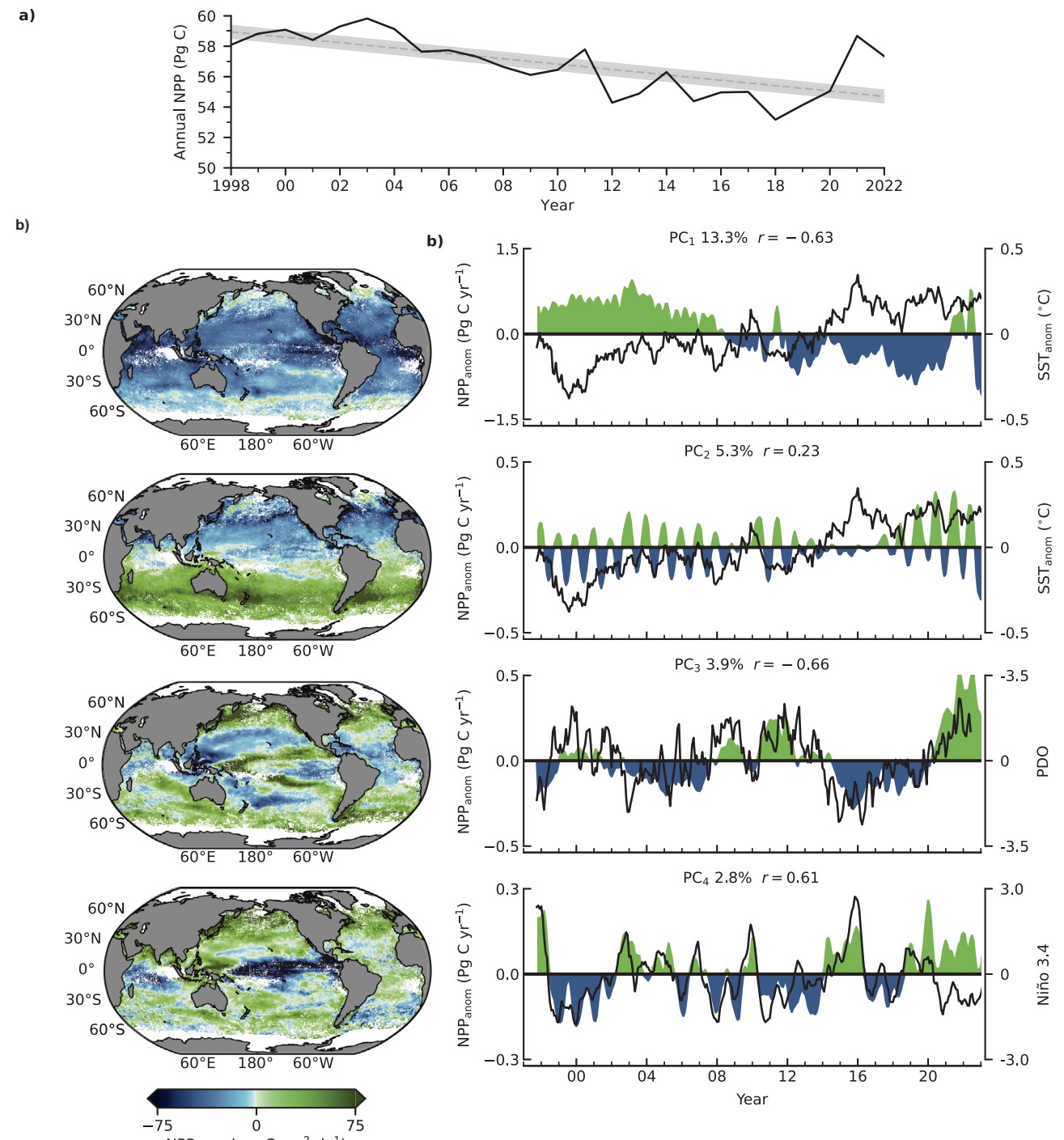

**Fig. 4 | Net Primary Production (NPP) variability. a** Global Annual NPP across the satellite record (solid line), the dashed line is the linear fit and the gray region is the 90% confidence intervals. **b** The spatial (left panels) and temporal (right panels) components of the four greatest modes of NPP anomaly variance. Modes of variance are labeled as PC$_{1\text{-}4}$ in the left panel and are shown in blue and green shading alongside their respective most correlated climate index (solid line, right axis). The percent explanatory variable of each mode is shown at the top of the panel alongside the correlation coefficient with the respective climate indices. Note that the magnitude of NPP anomalies through time have different scales.

photoacclimative response beneath the surface mixed layer that performs well compared to direct measurements[19] and, when applied globally, constitutes more than 25% of global annual NPP (Fig. 5). Declining absorption in the surface deepens euphotic depths, thus permitting greater NPP at depth (Supplementary Fig. 4). The fraction of NPP within the DCM is sensitive to the surface mixed layer depth (MLD) which is regulated by wind stress, buoyancy input, and other physical processes. The MLD is deepening in some regions of the ocean and becoming shallower in other regions (Supplementary Fig. 4), although in general, the density contrast across the base of the surface mixed layer is strengthening[32]. On aggregate, based on a photoacclimative response alone, NPP beneath the surface mixed layer has increased by almost 3 Pg C across the satellite record and now constitutes more than 30% of global NPP (Supplementary Fig. 4).

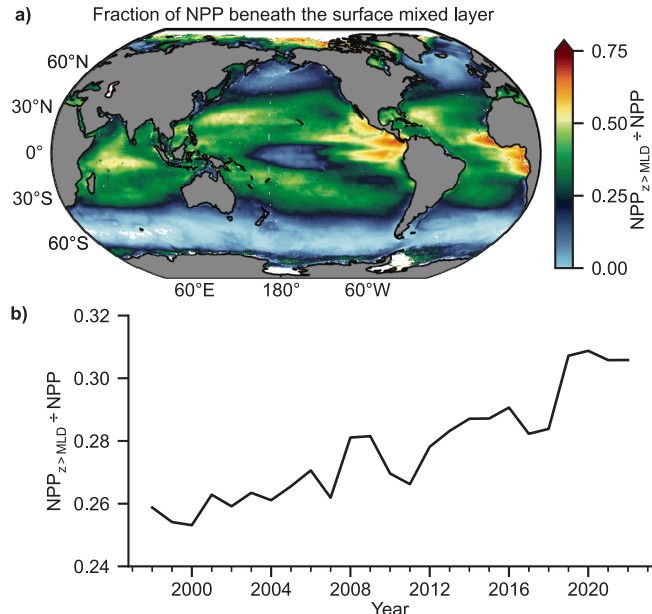

**Fig. 5 | Fractional contribution of Net Primary Production (NPP) beneath the surface mixed layer depth (MLD). a** 1997–2022 average NPP beneath the surface mixed layer depth ($NPP_{z>MLD}$). **b** The fractional contribution of $NPP_{z>MLD}$ to total NPP through time.

## Discussion

Oceanic waters are the predominant sink of anthropogenic heat, and the rate of SST warming is now accelerating[33]. The analyses presented here definitively show that NPP is decreasing across most of the stratified ocean. By segregating the atmospheric, bio-optical, and physiological components of NPP, our analysis further reveals that phytoplankton are absorbing less solar energy and have diminished growth rates, consistent with a physiological response to strengthening nutrient limitation that is emerging in a warming ocean. Given that this physiological response is primarily manifested as diminished photosynthetic light harvesting capacity, it is also likely that marine photosynthetic oxygen production is declining globally. This climate-mediated trend also has significant implications for global fisheries. Earth system models demonstrate that diminishing NPP in marine environments is amplified at higher trophic levels through the elongation of plankton food-webs caused by an increased dominance of smaller phytoplankton cells as well as declining zooplankton growth efficiency as consumers have limited energy above basal metabolic requirements[34,35].

The model and analyses presented here are agnostic to potential long-term changes in nutrient limitation beneath the surface mixed layer, and satellite-based NPP models that account for nutrient enhancement at depth use climatological nutrient data owing to the paucity of direct nutrient observations[27]. Given that the ocean color demonstrates that the warming and more stratified surface ocean is decreasing NPP in regions where DCMs are prevalent, and declining growth rates in these regions point to increasing nutrient limitation, the declining global NPP presented here may be conservative if nutrient limitation is becoming more prevalent beneath the surface mixed layer. Euphotic zone nutrient concentrations and NPP are declining in the subtropical North Atlantic[10], and this decline has been linked to weakening ventilation of nutrient enriched sub-tropical mode water from depth[36]. Global change models forecast that enhanced stratification will diminish nitrate concentrations and NPP throughout the euphotic zone[37], though this may be mitigated by enhanced atmospheric deposition in specific regions like the North Pacific[38]. In some regions, however, particularly equatorial waters,

nutrient enhancement beneath the surface mixed layer can fuel additional NPP[39].

Our analysis is observable within and consistent with reported changes in the satellite ocean color record[28], emphasizing the critical importance of continued investment in Earth-observing satellites and associated research. There are a number of emerging approaches and capabilities that can improve our ability to monitor changes in NPP at global scales[40]. For example, the now operational hyperspectral plankton, aerosol, cloud, and ecosystem mission can provide more accurate assessments of phytoplankton community composition and biomass[41]. NPP model variants that incorporate these features lead to more accurate and detailed assessments[40]. Moreover, phytoplankton exhibit intrinsic diel periodicity[42] that can be observed through geostationary Earth-observing satellites. Regional proof-of-concept studies have exploited this orbital class of satellites to observe diel changes in phytoplankton biomass to infer growth rates[43] and model NPP with estimates that compared favorably to in-situ measurements[44]. Looking beyond satellites, this study has shown that the global fraction of NPP beneath the surface mixed layer and beyond the reach of passive ocean color sensors is increasing. Thus, the growing importance of deep NPP demonstrates that additional measurement platforms (e.g., autonomous profiling floats), large-scale field surveys, and numerical models are essential to fully characterize and understand how climate change is impacting NPP and related global biogeochemical cycles.

## Methods
### Data sources
Absorption-based NPP is calculated using the CAFE model[19]. Level 3, monthly, 1/12° ocean color data (2022 reprocessing version) were downloaded from the NASA ocean color webpage for the SeaWiFS (09/1997–12/2007) and MODIS Aqua (07/2002–12/2022) missions. Inherent optical properties are determined via inversion of remote-sensing reflectance using the generalized inherent optical property model[18]. The CAFE model is spectrally explicit and evaluated at 10 nm increments in the visible spectrum (400–700 nm). Monthly MLD were retrieved from the Oregon state ocean productivity website (https://sites.science.oregonstate.edu/ocean.productivity/). The MLD is calculated as the depth where the density of water is $0.125\,kg\,m^{-3}$ greater than the density of near-surface water ($z = 10$ meters), where density is derived from HYCOM salinity and temperature data. Monthly 1/12° SST data are from the optimum interpolation sea surface temperature advanced very high resolution radiometer version 2.1[45].

### Satellite blending
To account for persistent biases between SeaWiFS and MODIS Aqua missions, we compared NPP from collocated monthly 1/12° data for the period 2002–2007. Overall, NPP showed remarkable correspondence between satellite missions (Supplementary Fig. 5); Type-1 linear regression returned an r value and slope of 0.96 and 0.97, respectively, and a mean absolute error of 1.09. That said, MODIS Aqua generally had higher NPP (Supplementary Fig. 5), therefore to accommodate for persistent spatial biases we followed the general convention of inter-satellite comparisons[17,46]: SeaWiFS data were adjusted (SeaWiFS*) by the monthly climatology differences derived from overlapping periods between the two missions. This adjustment resulted in an *r* value and slope of 0.98 and 0.99, respectively, and mean absolute error of 1.04. Satellite data (e.g., NPP, IOPs) were averaged between SeaWiFS* and MODIS-Aqua for the 2002–2007 overlap.

### Statistical and spatial analysis
For all time series, monthly anomalies were first calculated by subtracting monthly data from monthly climatology ($n = 303$). All statistics were performed using the Python package SciPy[47]. Spatially explicit trends and residuals were computed using ordinary least-

squares regression. Serial autocorrelation in anomaly residuals was quantified using a first-order model (i.e., monthly) that reduced the effective sample size of the time series. The standard error of the slope was recomputed using the reduced effective sample size. Uncertainties are expressed as 90% two-tailed confidence intervals derived from the effective sample size and recomputed standard error of the slope. EOF were derived after downsampling the data to an equal area $50 \times 50$ km grid. The full list of climate and oceanographic indices and their sources that were compared to the principal components are given in Supplementary Table 1. Gap filling of data using monthly climatological data was only performed to compute global NPP totals (Fig. 4a). On average less than 2% of ocean surface area where PAR > 1 mol photons $m^{-2}$ day$^{-1}$ required gap filling. All spatial analyses account for areal changes in pixel size and apply a global land mask created for satellite ocean color data[48].

## Data availability

Monthly global CAFE model output is available at https://doi.org/10.5281/zenodo.12586956. Remotely sensed data are available at https://oceancolor.gsfc.nasa.gov/l3/. Monthly global MLD data are available at http://orca.science.oregonstate.edu/2160.by.4320.monthly.hdf.mld030.hycom.php. Data generated in Figs. 1 through 5 are available at https://doi.org/10.5281/zenodo.15497141. All other data that support the findings of this study are available from the corresponding author upon reasonable request.

## Code availability

All major codes are available at doi.org/10.5281/zenodo.15497980. This repository includes Phyton and C code to compute NPP and associated data, and Python scripts for the trend and EOF analysis presented here.

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

## Acknowledgements
The authors would like to acknowledge funding from NASA grants 80NSSC21K0421, 80NSSC17K0560.

## Author contributions
G.M.S.: data analysis, visualization, original draft, review, and editing. J.F.: conceptualization, data analysis, review, and editing. T.K.W.: conceptualization, review and editing. K.H.H.: review and editing.

## Competing interests
The authors declare no competing interests.
