## [Peer Review file · Nature Communications]

Global declines in net primary production in the ocean color era

Corresponding Author: Dr Greg Silsbe

Version 0:

Reviewer comments:

Reviewer #1

(Remarks to the Author)

Using long-term spatiotemporal time series data of remote sensing ocean color, this study aims to show a long-term decline of NPP in the ocean, with clarification of spatial variation. The first reading gives the impression that this manuscript is very similar to Behrenfeld, et al. (1996), albeit with updated data. However, careful investigations of their analyses and results lead me to conclude that this work is interesting, particularly in providing a better mechanistic understanding of the global decline of NPP. My main concerns are about the presentation: 1. The novelty of this work is unclear due to the way of writing; as such, the manuscript reads like an update of Behrenfeld, et al. (1996) for none-specialists of ocean color remote sensing; 2. The methods are unclear. Below, I list some suggestions. My comments aim to make this manuscript accessible to a broad audience of Nat. Comm.

Main concerns:

1. The Abstract should include a better mechanistic explanation in addition to stating only the observed patterns. Otherwise, the findings seem similar to Behrenfeld et al. (1996), except that the time series is longer.
 2. I feel the Introduction needs substantial re-writing. The current version reads too much similar to Behrenfeld et al. (1996). The critical "knowledge gap" (the key innovation of this work) only starts to be unveiled from Line 61. Unfortunately, Lines 61 to 66 are too technical. Thus, the current writing does not successfully bring out the novelty of this work. I suggest some sentences in Lines 130-140 could be moved to the Introduction with some modifications. Importantly, readers expect an improved mechanistic understanding of the research topic.
 - I, by no means, suggest the authors criticize Behrenfeld et al. (1996). Nevertheless, the authors should make the sentences clear about what achievement is beyond Behrenfeld et al. (1996) in this work.
 3. The sentences in the Discussion read like implications of findings instead of a real discussion of the results.
 4. What I feel is missing in the Discussion is more detailed physiological components of remote sensing modeling. See a nice review paper: Westberry et al. (2023), actually by the same group of authors of this manuscript. I wish to see the authors comment on how the remote sensing modeling of NPP can be moved forward, for better mechanistic understanding.
 5. Fig. 2a. shows a very interesting reverse pattern from 2018 to 2021. Please discuss this.
 6. Extended Data Figures 3 and 4 in SI are more interesting than Figures 2 and 3 in the main text because Extended Data Figures 3 and 4 provide mechanistic explanations of decreasing NPP.
- The spatial pattern of NPP (PC 3-4) is, to a large extent, affected by changes in circulation and SST that are driven by climate. Thus, the correlation between the PC3-4 and the climate index is not surprising. I feel detailed explanations of Figure 2 are not the most novel findings of this work. Instead, I wish to see more detailed explanations and discussion for Extended Data Figures 3 and 4, which provide "mechanistic understanding from the viewpoint of pigment absorption."
7. Monthly anomalies were calculated by subtracting monthly data from monthly climatology. According to Fig 2b, PC2, the seasonality remains very visible. This issue is probably not critical to the key findings of this work. However, please discuss

this, to avoid confusion.

8. Methods: How to calculate and obtain Extend Data Figures 2 to 5 need detailed explanations in the Method section. Codes to reproduce the results should be provided along with the paper. The majority of readers are not experts in remote sensing.

9. In many places, the wording is too technical, as indicated in the editorial suggestions below. Please clarify the biological meanings of those. I just made some examples; the authors should carefully revise the manuscript.

Editorial suggestions:

Line 28: Please clarify "phytoplankton pigmentation"?

Line 33: Please explain the biological meaning of "declining pigment absorption". I do not think the broad audience of Nat. Comm. understand this term.

Finally

I look forward to using this as teaching material in my Biological Oceanography class.

Chih-hao Hsieh

Reviewer #2

(Remarks to the Author)

Review of Silsbe et al. "Global declines in net phytoplankton production in the ocean color era"

This manuscript presents an analysis of a merged 25 year dataset of net primary production derived from satellite data. The results suggest a decline in NPP in low latitude regions, and furthermore explores the photo-physiological drivers underlying the trends. The manuscript presents some interesting and relevant results, but I would urge the authors to assess the uncertainties introduced by their choice of satellite time series data and how the merged time series is created.

Methods: It would be wise to check the merged SeaWiFS-MODIS-Aqua time series the authors have created for any discontinuities introduced when different combinations of satellites are included in the time series. Change point detection is one option. Existing packages are available in matlab as findchangepts or R as changepoint (and I'm sure Python has something equivalent). If the algorithms pick out dates associated with when an additional data source is introduced/removed, then artificial trends could be created in the time series.

I also wonder why the authors didn't use the OC-CCI dataset, as it incorporates multiple ocean colour satellites (not just the 2 used here) and has made a huge effort in producing a consistent, climate record-quality timeseries. That would overcome my misgivings about using the 2-satellite merged time series here. There has also been reported deterioration in the MODIS-Aqua dataset for the last couple of years due to sensor degradation. Has this been taken into account here?

Given the lack of data at high latitudes in winter, due to both cloud/ice cover and low sun angle, the gap-filling routine applied here must be effectively "making up" a significant fraction of the time series at high latitudes. Some discussion of the implications of this for your results needs to be added.

Lines 26-27: the authors use "net phytoplankton production (NPP)" here rather than the more common "net primary production" (also commonly abbreviated as NPP). Is that intentional?

Line 104: "a 7.4% over" – this is incomplete.

Lines 115-119: the second mode of variance appears to represent the opposed seasonal cycles of the Southern and Northern Hemispheres, and indeed I think I counted 1 peak/trough per year of the time series too. The authors hoped to remove the seasonal cycle by taking anomalies from the climatological year, but there seems to be a large amount of residual seasonality in the time series. The authors should consider whether there is a better way to remove the seasonal cycle so it isn't so dominant in the EOF analysis. Could they try applying an EOF to the full time series (before seasonality is removed), and the 1st mode will probably be the seasonal signal, then subtract that from the full time series, and run the EOF again??

Lines 120-124: the aseasonal pattern of the Southern Ocean is emphasised here, but the EOF spatial pattern is as strong in the Northern Hemisphere as it is in the SOcean. The discussion needs to be more balanced here.

Line 135: define ϕ_{μ} again here to help the reader

Lines 145-147: is the result reported in this sentence shown in this manuscript (if so, direct the reader to a figure) or is it from literature and so needs a reference?

Line 171: how is the SML defined in this study?

Lines 188: specify that the NPP models used here are satellite data-based

Line 206: I find use of the word 'evolution' here potentially misleading as the authors aren't really talking about an evolutionary process (I think? Given the 25 year time scale of the study)

Figure 1a: mark the non-significant pixels on the map with hatching (or something)

Figure 1c: it's very hard to distinguish between non-significant and small trends with this colour scheme. As for figure 1a, add hatching to identify non-significant pixels.

Figure 1d: specify that the inset is the SST and what timeframe it covers

Figure 1e: I didn't really understand this figure! What are the 'top' and 'bottom' sections of the plot. Why is the shaded area both positive and negative?

Lines 402-403: please acknowledge here the lack of data in winter at high latitudes, and that a lot of gap-filling will need to have taken place to make a complete timeseries.

Extended Figure 2c: I didn't follow what "spatially averaged latitude of austral waters" meant

Extended Figure 3d-f: are these panels needed? They're not very informative.

Extended Figure 3g-i: add to caption that the legend also includes the proportion of the ocean in each category.

Version 1:

Reviewer comments:

Reviewer #1

(Remarks to the Author)

My comments have been well addressed. The revision is satisfactory.

Reviewer #2

(Remarks to the Author)

I find the manuscript improved in terms of clarity for the reader and outlining the implications of the analysis. I just have a couple of minor corrections:

- Title still includes "net phytoplankton production" instead of "net primary production", although it has been changed in the rest of the manuscript.
- Line 186 should refer to Figure 5 (currently says Figure 4).
- Line 196: anthropogenic heat is not predominantly being absorbed by ecosystems, but rather by the ocean water itself. Of course there will be effects on the ecosystem, but it is wrong to say that "marine ecosystems are the predominant sink of anthropogenic heat".

REVIEWER COMMENTS

Reviewer #1 (Remarks to the Author):

Using long-term spatiotemporal time series data of remote sensing ocean color, this study aims to show a long-term decline of NPP in the ocean, with clarification of spatial variation. The first reading gives the impression that this manuscript is very similar to Behrenfeld, et al. (1996), albeit with updated data. However, careful investigations of their analyses and results lead me to conclude that this work is interesting, particularly in providing a better mechanistic understanding of the global decline of NPP. My main concerns are about the presentation: 1. The novelty of this work is unclear due to the way of writing; as such, the manuscript reads like an update of Behrenfeld, et al. (1996) for none- specialists of ocean color remote sensing; 2. The methods are unclear. Below, I list some suggestions. My comments aim to make this manuscript accessible to a broad audience of Nat. Comm.

Main concerns:

1. The Abstract should include a better mechanistic explanation in addition to stating only the observed patterns. Otherwise, the findings seem similar to Behrenfeld et al. (1996), except that the time series is longer.

The abstract has been revised with a better mechanistic explanation (Lines 31:34, please note that Lines refer to the manuscript where track changes are hidden).

2. I feel the Introduction needs substantial re-writing. The current version reads too much similar to Behrenfeld et al. (1996). The critical "knowledge gap" (the key innovation of this work) only starts to be unveiled from Line 61. Unfortunately, Lines 61 to 66 are too technical. Thus, the current writing does not successfully bring out the novelty of this work. I suggest some sentences in Lines 130-140 could be moved to the Introduction with some modifications. Importantly, readers expect an improved mechanistic understanding of the research topic. I, by no means, suggest the authors criticize Behrenfeld et al. (1996). Nevertheless, the authors should make the sentences clear about what achievement is beyond Behrenfeld et al. (1996) in this work.

The introduction has been substantially revised. The critical knowledge gap is now more explicitly stated with less technical jargon (Line 60-74). Following your suggestion, lines from the Results have been incorporated into this new text. The critical knowledge gap still begins in the third paragraph, as we feel it is still important to introduce NPP (Paragraph 1) and remote sensing (Paragraph 2) to a general audience before arriving at this text.

3. The sentences in the Discussion read like implications of findings instead of a real discussion of the results.

The Discussion has been revised (see next comment).

4. What I feel is missing in the Discussion is more detailed physiological components of remote sensing modeling. See a nice review paper: Westberry et al. (2023), actually by the same group of authors of this manuscript. I wish to see the authors comment on how the remote sensing modeling of NPP can be moved forward, for better mechanistic understanding.

We have added several sentences summarizing some of the key concepts how remote sensing modeling can be moved forward, as also discussed in Westberry et al. (2023) (Lines 225:238).

5. Fig. 2a. shows a very interesting reverse pattern from 2018 to 2021. Please discuss this.

Thank you for this comment. We cannot fully explain the complete reversal pattern, however some of this reversal is due to the transition from a strongly positive to strongly negative PDO (Lines 173-174).

6. Extended Data Figures 3 and 4 in SI are more interesting than Figures 2 and 3 in the main text because Extended Data Figures 3 and 4 provide mechanistic explanations of decreasing NPP.

The spatial pattern of NPP (PC 3-4) is, to a large extent, affected by changes in circulation and SST that are driven by climate. Thus, the correlation between the PC3-4 and the climate index is not surprising. I feel detailed explanations of Figure 2 are not the most novel findings of this work. Instead, I wish to see more detailed explanations and discussion for Extended Data Figures 3 and 4, which provide “mechanistic understanding from the viewpoint of pigment absorption.”

Consistent with changes to the abstract and introduction, we have brought forth the most pertinent data from the Extended Data Figures 3 and 4 to form a new figure (Figure 3) in the main text and changed the text accordingly, with commensurate changes in Extended Data Figures.

7. Monthly anomalies were calculated by subtracting monthly data from monthly climatology. According to Fig 2b, PC2, the seasonality remains very visible. This issue is probably not critical to the key findings of this work. However, please discuss this, to avoid confusion.

Thank you for this comment, given a similar comment from Reviewer #2 we have strived to provide a clearer and more balanced analysis of PC2. Here we briefly summarize this change, please see Lines 161-169 and the newer Extended Data Figure (now Extended Data Figure 4) that provides a clearer explanation of PC2. While the PC2 time-series appears seasonal, it in fact represents regions of the ocean where summer NPP is decreasing but winter productivity is increasing, leading to an overall shift towards more muted seasonality. This phenological shift is simultaneously occurring in both hemispheres, such that declining summer NPP in one hemisphere is offset by increases in winter NPP in the other hemisphere and vice-versa, a phenomenon that gives rise to the pseudo-seasonal time series and strong latitudinal gradients shown in Figure 4. That this phenomenon constitutes 5.3% of the global NPP anomaly variance is notable as it exceeds the variance imparted by both the PDO and ENSO.

8. Methods: How to calculate and obtain Extend Data Figures 2 to 5 need detailed explanations in the Method section. Codes to reproduce the results should be provided along with the paper. The majority of readers are not experts in remote sensing.

Trend analyses are identical for NPP and all other satellite data. We have attempted to make this more clear in the Results (Lines 125-127) as well as the Methods. We have created a GitHub repository with all relevant code (Line 444-446).

9. In many places, the wording is too technical, as indicated in the editorial suggestions below. Please clarify the biological meanings of those. I just made some examples; the authors should carefully revise the manuscript.

Editorial suggestions:

Line 28: Please clarify "phytoplankton pigmentation"?

This has been changed simply to 'light-harvesting pigment capacity' in our revised abstract (Line 33). We hope the inclusion of light-harvesting capacity is enough for a general audience to understand the physiological role of pigments.

Line 33: Please explain the biological meaning of "declining pigment absorption". I do not think the broad audience of Nat. Comm. understand this term.
See previous comment.

Finally

I look forward to using this as teaching material in my Biological Oceanography class.

Thank you! We strive for effective science visualization.

Reviewer #2 (Remarks to the Author):

Review of Silsbe et al. "Global declines in net phytoplankton production in the ocean color era"

1. This manuscript presents an analysis of a merged 25 year dataset of net primary production derived from satellite data. The results suggest a decline in NPP in low latitude regions, and furthermore explores the photo-physiological drivers underlying the trends. The manuscript presents some interesting and relevant results, but I would urge the authors to assess the uncertainties introduced by their choice of satellite time series data and how the merged time series is created.

2. Methods: It would be wise to check the merged SeaWiFS-MODIS-Aqua time series the authors have created for any discontinuities introduced when different combinations of satellites are included in the time series. Change point detection is one option. Existing packages are available in matlab as findchangepts or R as changepoint (and I'm sure Python has something equivalent). If the algorithms pick out dates associated with when an additional data source is introduced/removed, then artificial trends could be created in the time series.

We conducted a change point analysis of global NPP anomalies: Autoregression in the anomalies was removed as described in the methods, then multiple change point detection was determined using the Python package 'ruptures' (C. Truong et al. 2020) parameterized with a least square deviation cost function and a minimum length between change points of 3 years (results were robust across different cost functions and lengths). This analysis identified 3 change points (Jun-2004, Jul-2011, Jun-2019) that are different than the introduction of MODIS Aqua (Aug-2002) and our discontinuation of SeaWiFS (Dec-2008).

3. I also wonder why the authors didn't use the OC-CCI dataset, as it incorporates multiple ocean colour satellites (not just the 2 used here) and has made a huge effort in producing a consistent, climate record-quality timeseries. That would overcome my misgivings about using the 2-satellite merged time series here.

We considered the OC-CCI dataset but decided upon exclusively SeaWiFS and Aqua for four reasons. 1: The OC-CCI does not have a PAR product that is essential to all NPP models. 2: The MODIS Aqua record, that dominates the time series (20 years), has greater spectral resolution than OC-CCI. 3: The spectral inversion algorithm employed by the OC-CCI (Lee et al. 2009) has higher uncertainty (Brewin et al. 2015, <http://dx.doi.org/10.1016/j.rse.2013.09.016>) than the NASA operational GIOP spectral inversion model (Werdell et al. 2013) employed here. 4: At time of submission, the OC-CCI was using an older reprocessing version (R2018.0) of NASA ocean color data. More to the point, our methods essentially follow the bias correction method adopted by the OC-CCI. We thank you for the comment and to allay future misgivings, we have made this more clear in our Methods (Line 413-414, please note that Lines refer to the manuscript where track changes are hidden).

4. There has also been reported deterioration in the MODIS-Aqua dataset for the last couple of years due to sensor degradation. Has this been taken into account here?

As we were aware of the deterioration on MODIS-Aqua, we purposefully ended our time series in 2022 to minimize instrument artefacts and to align with the most recent MODIS Aqua reprocessing at time of submission (R2022.0, <https://oceancolor.gsfc.nasa.gov/data/reprocessing/r2022/>).

5. Given the lack of data at high latitudes in winter, due to both cloud/ice cover and low sun angle, the gap-filling routine applied here must be effectively "making up" a significant fraction of

the time series at high latitudes. Some discussion of the implications of this for your results needs to be added.

We have clarified this in our methods to more clearly state that gap filling was only performed to compute the global NPP Trend (Figure 4A), and that the missing area constituted less than 2% of the ocean surface where PAR > 1 mol photons m⁻² day⁻¹. In other words once you factor in areas subjected to polar night when NPP is negligible, gap filling is in fact minor. (Lines 429-433).

6. Lines 26-27: the authors use “net phytoplankton production (NPP)” here rather than the more common “net primary production” (also commonly abbreviated as NPP). Is that intentional?

We have changed NPP to net primary production, and when introducing NPP the first time we state ‘phytoplankton Net Primary Production (NPP)’. We originally used the term phytoplankton, and add the distinction above, as non-phytoplankton autotrophs (i.e. kelps, sea grasses) contribute a very small fraction of marine primary production.

7. Line 104: “a 7.4% over” – this is incomplete.

The sentence now states ‘a 7.4% reduction over the satellite record’ Line 147-148.

8. Lines 115-119: the second mode of variance appears to represent the opposed seasonal cycles of the Southern and Northern Hemispheres, and indeed I think I counted 1 peak/trough per year of the time series too. The authors hoped to remove the seasonal cycle by taking anomalies from the climatological year, but there seems to be a large amount of residual seasonality in the time series. The authors should consider whether there is a better way to remove the seasonal cycle so it isn’t so dominant in the EOF analysis. Could they try applying an EOF to the full time series (before seasonality is removed), and the 1st mode will probably be the seasonal signal, then subtract that from the full time series, and run the EOF again?? Lines 120-124: the aseasonal pattern of the Southern Ocean is emphasised here, but the EOF spatial pattern is as strong in the Northern Hemisphere as it is in the SOcean. The discussion needs to be more balanced here.

Thank you for this comment, given a similar comment from Reviewer #12 we have strived to provide a clearer and more balanced analysis of PC2. Here we briefly summarize this change, please see Lines 161-167 and the newer Extended Data Figure (now Extended Data Figure 4) that provides a clearer explanation of PC2. While the PC2 time-series appears seasonal, it in fact represents regions of the ocean where summer NPP is decreasing but winter productivity is increasing, leading to an overall shift towards more muted seasonality. This phenological shift is simultaneously occurring in both hemispheres, such that declining summer NPP in one hemisphere is offset by increases in winter NPP in the other hemisphere and vice-versa, a phenomenon that gives rise to the pseudo-seasonal time series and strong latitudinal gradients shown in Figure 4. That this phenomenon constitutes 5.3% of the global NPP anomaly variance is notable as it exceeds the variance imparted by both the PDO and ENSO.

9. Line 135: define phi_mu again here to help the reader

Following the suggestion of reviewer 1, this sentence has been moved to the introduction and immediately follows phi_mu being defined (Line 65-66).

10. Lines 145-147: is the result reported in this sentence shown in this manuscript (if so, direct the reader to a figure) or is it from literature and so needs a reference?

Originally, this sentence was missing a reference to an extended data figure. Following comments by Reviewer 1, this figure is now in the main section of the manuscript and the text has been modified. (Lines 123-129).

11. Line 171: how is the SML defined in this study?

The definition of the surface mixed layer is provided in the Methods (Lines 405-406).

12. Lines 188: specify that the NPP models used here are satellite data-based.

The line now reads 'and satellite-based NPP models' (Line 192).

13. Line 206: I find use of the word 'evolution' here potentially misleading as the authors aren't really talking about an evolutionary process (I think? Given the 25 year time scale of the study)' oxygen evolution' has been changed to 'oxygen production' (Line 213).

Figure 1a: mark the non-significant pixels on the map with hatching (or something)

Hatching has been added to all relevant figures.

Figure 1c: it's very hard to distinguish between non-significant and small trends with this colour scheme. As for figure 1a, add hatching to identify non-significant pixels.

Hatching has been added to all relevant figures.

Figure 1d: specify that the inset is the SST and what timeframe it covers

We have added an inset label.

Figure 1e: I didn't really understand this figure! What are the 'top' and 'bottom' sections of the plot. Why is the shaded area both positive and negative?

In the original version, the y-axis was mirrored to show how much of the ocean, binned by average SST, had significant increasing NPP (top half), or decreasing NPP (bottom half). We have simplified this figure to show areas of significant NPP trends regardless of direction and changed the figure caption accordingly.

Lines 402-403: please acknowledge here the lack of data in winter at high latitudes, and that a lot of gap-filling will need to have taken place to make a complete timeseries.

Extended Figure 2c: I didn't follow what "spatially averaged latitude of austral waters" meant

This section has changed following both reviewers suggestion, and the sentence no longer appears.

Extended Figure 3d-f: are these panels needed? They're not very informative.

This figure has been changed, and following comments from Reviewer #1, is now in Figure 3.

Extended Figure 3g-i: add to caption that the legend also includes the proportion of the ocean in each category.

Extended Data Figure 2 Caption now reads 'Frequency map showing the largest statistically significant normalized trend (increasing or decreasing) for inherent optical properties.'

REFERENCES

C. Truong, L. Oudre, N. Vayatis. Selective review of offline change point detection methods. *Signal Processing*, 167:107299, 2020

Response To Reviewers

Reviewer 1: No revisions given.

R2: I just have a couple of minor corrections:

- Title still includes "net phytoplankton production" instead of "net primary production", although it has been changed in the rest of the manuscript.

Title Changed.

- Line 186 should refer to Figure 5 (currently says Figure 4).

Correction made (now Line 187).

- Line 196: anthropogenic heat is not predominantly being absorbed by ecosystems, but rather by the ocean water itself. Of course there will be effects on the ecosystem, but it is wrong to say that "marine ecosystems are the predominant sink of anthropogenic heat".

'Marine Ecosystems' replace with 'Oceanic waters' (now Line 197).

END